# The Effect of Hydrological Connectivity on the Zooplankton Structure in Floodplain Lakes of a Regulated Large River (the Lower Vistula, Poland)

**Paweł Napiórkowski *, Martyna Bąkowska, Natalia Mrozińska, Monika Szymańska, Nikola Kolarova and Krystian Obolewski**

Department of Hydrobiology, Faculty of Natural Sciences, Kazimierz Wielki University, 85-064 Bydgoszcz, Poland; bakowska@ukw.edu.pl (M.B.); nattam@ukw.edu.pl (N.M.); m.szymanska9306@gmail.com (M.S.); n.kolar77@gmail.com (N.K.); obolewsk@ukw.edu.pl (K.O.)
* Correspondence: pnapiork@ukw.edu.pl; Tel.: +48-523-419-171

**Abstract:** The zooplankton community structure and diversity were analysed against the gradient of floodplain lakes connectivity and water level under different flood-pulse dynamics in the Vistula River. The lakes differed in terms of hydrology, among others in the degree/type of their connection with the river (permanent, temporary and no connection). The study was conducted during the growing seasons in the years 2006–2013 and involved the lower Vistula River and three floodplain lakes: isolated, transitional and connected. Water samples were collected biweekly from April to September. Zooplankton was the most diverse and abundant in the transitional lake (the highest Shannon $\alpha$-diversity index H' and Pielou's evenness index J'). The gentle washing of the lakes might have stimulated the development of zooplankton in accordance with the Intermediate Disturbance Hypothesis. The diversity and density of zooplankton were higher in the connected lake compared to the isolated one. We confirmed the hypothesis that zooplankton should be more abundant and diverse in floodplain lakes connected with the river (or transitional) than in isolated ones. Zooplankton analyses indicated that hydrological conditions (flood-pulse regime) contributed most substantially to zooplankton diversity and density in the floodplain lakes of the lower Vistula valley.

**Keywords:** invertebrates; hydrological regime; diversity; water bodies

## 1. Introduction

Floodplain lakes, known for high biodiversity and ecological value, are important elements of landscapes with large rivers [1–4]. A floodplain lake is defined as a small water body located in a river valley, permanently/periodically connected to or isolated from the main river channel, formed when a meander is cut off naturally or separated from the river by a flood embankment. Parts of the old river channel connected periodically or permanently with the river bed are also considered as floodplain lakes [5]. Floodplain lakes are generally shallow, astatic water bodies with varying environmental conditions and macrophyte dominance. They can be seen as flowing-to-stagnant water transition zones [6–8].

Floodplain lakes can be divided into isolated from the river, temporarily connected to the river, or permanently connected to the river [5]. According to the flood-pulse concept [9] the functioning of floodplain lakes depends on periodic river flooding. The resulting connection between the river and the lake allows the exchange of water with nutrients and organisms between all elements of the river system [1,2,10,11]. Hydrological connectivity is observed on four levels of fluvial systems: longitudinal, lateral, vertical and temporal [2]. The lateral and temporal (long-term dynamics) connectivity were investigated.

The degree of hydrological connectivity, duration and frequency of flooding as well as lake water level depend on many factors such as lake location, its distance from the river, river level fluctuations and catchment area [2,12]. Increased hydrological connectivity or flood-pulse act as a homogenizing factor [13,14].

Dias et al. [15], Dittrich et al. [16], Anderson and Bonecker [17] and Schöll et al. [18] point out that plankton communities in floodplain lakes are determined primarily by these two factors: whether a lake is connected with the river and whether inundation causes any disturbances in a lake.

The intensity of flooding determines species diversity and density in floodplain lakes. According to the Intermediate Disturbance Hypothesis (IDH) [19], medium flooding can increase species diversity [20]. The IDH predicts low species diversity in ecosystems exposed to high and low levels of disturbance. Under these conditions the survival is guaranteed only for species which can easily adapt to changing and/or extreme conditions or can quickly recolonize a given ecosystem (e.g., floodplain lakes after floods).

There are two theories describing how the connection between a floodplain lake and a river affects the zooplankton community. Kobayashi et al. [21] and Lemke et al. [22] observed high zooplankton density in hydrologically isolated floodplain lakes. The inflow of river water to lakes connected with the main river channel can periodically destabilize their environmental conditions. Lower water temperature and transparency inhibit the growth of macrophytes, which normally provide a habitat for zooplankton (space and food). Conditions in isolated floodplain lakes will lead to greater diversity and abundance of planktonic crustaceans (more species and higher abundance). However, Hein et al. [23] and Kasten [24] hypothesize that zooplankton should be more abundant and diverse in lakes connected to the river than in the isolated ones.

The main objective of our study was to answer the question of how different hydrological connectivity between a large regulated river and its floodplain lakes can shape zooplankton communities. Before the investigation we put forward the following hypotheses: (1) The river zooplankton will be less diverse and less abundant than zooplankton in the studied floodplain lakes owing to specific environmental conditions in the river (turbulent water flow, lower temperature, etc.). (2) The degree of connectivity between a particular lake and the river will affect the zooplankton structure. The diversity and abundance of zooplankton will be lower in the lakes connected with the river than in the isolated one as a result of less stable environmental conditions in the former. (3) Flood-pulse dynamics will have an impact on a degree of connectivity between the lakes and the river and will affect the zooplankton structure. During high water level the predominance of small organisms (rotifers) will be observed. During low water level the abundance of crustaceans will increase.

## 2. Materials and Methods

The study involved three floodplain lakes lying in the lower Vistula valley. Over almost its entire length Vistula is a typical lowland river. The first floodplain terrace has many lakes which are the remnants of the Vistula backwaters and are periodically flooded. The investigated lakes were created after the construction of flood embankments during the river regulation in the 19th century [25]. The lakes are shallow and relatively young (ca. 150 years). Before regulation the Vistula was a braided river, so its old riverbeds tend to have an elongated shape and be half-open (semi-lotic) or closed (lenitic) (Figure 1).

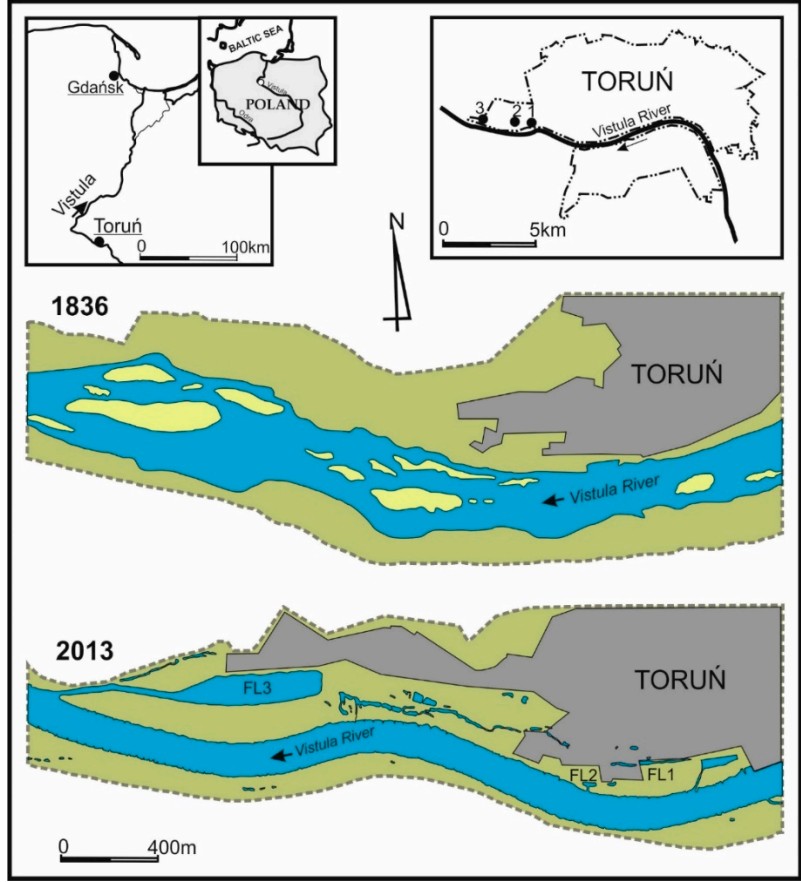

**Figure 1.** Hydrological system of the lower Vistula River in Toruń (Poland) before regulation (1836) and today (2013). FL1—floodplain lake isolated from the river, FL2—floodplain lake periodically connected to the river (transitional), FL3—floodplain lake permanently connected to the river.

The three studied floodplain lakes of the lower Vistula River are located in the city of Toruń: FL1 (53°00′ N, 18°34′ E)—isolated from the river (*n* = 34), FL2 (53°00′ N, 18°33′ E)—temporarily isolated from the river (*n* = 38: 19 with connection and 19 without connection) and FL3 (53°01′ N, 18°30′ E)— permanently connected with the river by a long channel (*n* = 39) (*n*—number of samples). FL1, located in Toruń city park at the 737th km of the river, is a small (2.5 ha) and shallow (2.0 m) water body without a direct surface contact with the Vistula River. It has a submerged macrophyte community and a rich littoral zone. The following macrophyte species inhabit the lake: yellow pond-lily (*Nuphar lutea*), rigid hornwort (*Ceratophyllum demersum*), star duckweed (*Lemna trisulca*). Different species of filamentous algae are also found in the lake.

FL2, located in Toruń at the 738th km of the river is small (1.0 ha). At the medium water level in the Vistula it is connected with the river through a wide channel (up to 30 m). At the low water level the lake is isolated. The area at the northern shore is covered by gardens. With a small surface and low exposure to wind, the lake is not very susceptible to water mixing. At the low water level in the Vistula elodeids (mainly *C. demersum* and *Potamogeton* spp.) could have developed however their density was relatively small (no reeds observed).

FL3 is located at the 745th km of the Vistula, (the largest) (Figure 1, Table 1). It is permanently connected with the river through a 1.2 km channel, constructed to make possible timber transportation. The direct catchment comprises woodlands, grasslands and agricultural areas. During growing seasons we observed rapid growth of elodeids on the lake bottom, resulting from the low water level in the Vistula. The dominant species included Canadian waterweed (*Elodea canadiensis*), hornwort (*C. demersum*), and potamogetons (*Potamogeton* spp.). At very low water level submerged vegetation

filled almost the entire water column. Although floodplain lakes differed in size and morphology, the habitat complexity during the vegetation season seemed to be low.

**Table 1.** Lateral hydrological connectivity definitions for floodplain lakes of the lower Vistula River. Type corresponds to categories of hydrological connectivity (HC). Depth is at the deepest part of the floodplain lakes in 2006–2013. Age is the time since the cutoff of floodplain lakes, or construction, depending on origin.

| Site | Depth (m) | Age (yr) | Area (ha) | Length (m) | Geographic Coordination | HC (%) | HC Definition |
|------|-----------|----------|-----------|------------|-------------------------|--------|---------------|
| VR | - | - | - | - | N 53°00′ E 18°60′ | - | Vistula River |
| FL1 | 2.5 | 155 | 2.8 | 640 | N 53°00′ E 18°34′ | 0 | Floodplain lake without connection with the Vistula River |
| FL2 | 0.6–1.6 | 155 | 1.0 | 160 | N 53°00′ E 18°33′ | 50 | Floodplain temporarily connected with the Vistula River |
| FL3 | 11.0 | 155 | 71 | 1800 | N 53°01′ E 18°30′ | 100 | Floodplain permanently connected with the Vistula River by one arm 1.2 km length and width 50 m |

Our studies were conducted during the growing seasons in the years 2006–2013 and involved the lower Vistula River and three floodplain lakes. Water samples were collected biweekly from April to September. A total of 135 samples were collected. Water samples were collected with a 1 L Patalas bucket at the depth of ca. 0.5 m in the central part of each water body. Water was filtered through a plankton net, mesh size 25 μm. Ten litres of water were filtered to obtain one sample of zooplankton. All zooplankton samples were preserved in Lugol's solution [26,27]. The sample volume (10 L) was adjusted to 10 mL and a 1 mL aliquot of well mixed concentrate was pipetted into a Segdwick–Rafter chamber. The zooplankton was counted under a microscope in a Segdwick–Rafter chamber by the sub-sample method [26]. The abundance of zooplankton was calculated per volume of 1 L of water. The identification of zooplankton was performed with the use of a light microscope Nikon Alphaphot-2, a camera and MultiScan computer software for image analysis. The taxonomical identification of zooplankton was based on the commonly available studies and keys [26,28–33].

To characterize the abundance-dominance relationship the Shannon $\alpha$-diversity index (H′) and Pielou evenness index (J′) were used. Sampling was accompanied by the measurements of physical and chemical parameters of water, such as: Secchi disk visibility (SD, m) (except the river), temperature ($T_w$, °C), dissolved oxygen (DO, mg $L^{-1}$), conductivity (EC, μS $cm^{-1}$), and pH. Measurements of physicochemical parameters were performed with the use of multimeter WTW Multi 3430SET F (Xylem Analytics, Weilheim, Germany) field probes. Data on the water level (WL, cm) in the Vistula River in Toruń city were obtained from the Meteorological and Hydrological Institute, the Regional Hydrological and Meteorological Station in Toruń.

For statistical analysis, the investigated floodplain lakes were classified based on the degree of their connectivity with the river. The final dataset contained basic environmental variables: dates of analysis combined with the degree of connectivity between each lake and the river (1—isolated, 2—transitional; 3—connected), water level, water physicochemical parameters, zooplankton richness and density.

Similarity analysis performed to classify abiotic data and confirm floodplain lake types was based on non-metric multidimensional scaling (nMDS) using Bray–Curtis dissimilarity indices and software Past v.3.01 [34].

To evaluate general differences in the zooplankton structure we investigated three water bodies (FL1, FL2 and FL3) on the given dates and performed analysis of variance (ANOVA) with the Kruskal–Wallis test (*K–W*) followed by post-hoc Dunn's test in Prism 5.01 software (GraphPad Software, San Diego, CA, USA).

The following variables were determined: water level, water physicochemical parameters ($T_w$, SD, pH, DO, EC), species richness (zooplankton species richness, rotifer species richness and crustacean species richness) and mean zooplankton density (total zooplankton density, rotifer, crustacean and predominant species densities), α-diversity and evenness indexes.

Redundancy analysis (RDA) was performed using the covariance method to determine the relative significance of environmental factors in explaining the variability of the tested samples. The dataset was log transformed (log ($n$ + 1)) and centred on species, as this was obligatory for the constrained linear methods.

Consequently, the relationships between predictors (dates, sites, water level, water physicochemical properties) and zooplankton density were analysed by RDA [35,36]. The statistical significance of canonical axes was determined in the Monte Carlo permutation test [37]. A subset of independent variables representing the relationships between environmental factors and the taxa of planktonic fauna was identified by eliminating factors that were not significant for the model. The data were processed statistically in Canoco 5.0 software (Wageningen University & Research, Wageningen, Holland) at probability levels of * $p < 0.05$, ** $p < 0.01$ and *** $p < 0.001$ [38].

To explore significant positive and negative relationships between zooplankton and specified descriptors of physicochemical properties of lake water, *t*-value biplots (with van Dobben circles), which approximate the *t*-values of the regression coefficients of a weighted multiple regression, were generated. The *t*-value biplots indicated the zooplankton data that to a large extent reacted to the tested factor [38].

## 3. Results

### 3.1. Hydrological Conditions

From 2006 to 2013 water levels in the Vistula River at a gauging station in Toruń ranged from 1.45 m to 8.36 m above sea level (a.s.l.) and the average level was 3.0 m a.s.l. (Figure 2). In the research period water levels ranged from 1.45 m to 5.25 m a.s.l., with the average value of 2.84 m a.s.l.

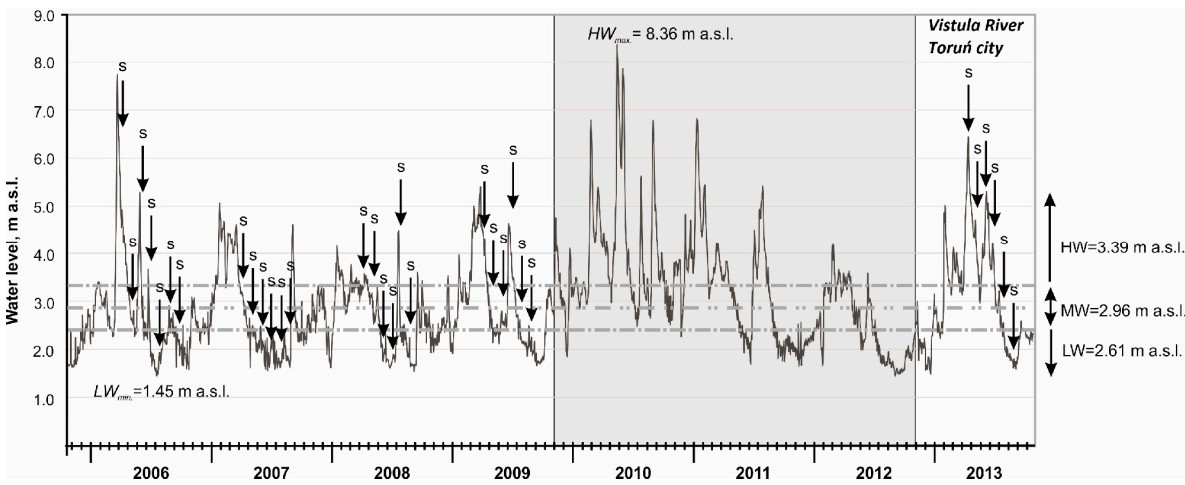

**Figure 2.** Hydrograph of the lower Vistula River from the gauge station in Toruń. $HW_{max}$—the maximum water level, $LW_{min}$—the lowest water level. HW—average of high water levels, MW—average of mean water levels, LW—average of low water levels have been calculated from the study period 2006–2009 and 2013 (grey area indicates a period without sampling). Arrows with "s" indicate sampling times.

The majority of samples (2006–2009) were collected during low and average water levels in the Vistula River. Only in 2013, the river level was high during sampling. In May 2010, a catastrophic flood that occurred on the Vistula inundated all the investigated floodplain lakes and changed their

biotic and abiotic conditions. To eliminate the impact of the flood on the results, samples were not collected from 2010 to 2012. In 2013 we assumed that the river had returned to the hydrological state that was observed before the flood.

FL3 is directly connected with the river channel, therefore changes in lake levels and discharges are determined by the river. FL2 did not have permanent surface connectivity with the river during the research period. When the water level in the Vistula dropped below 2.31 m a.s.l. no connectivity was observed. For 38 observations conducted in the transitional floodplain lake, half were made during the isolation period. FL1, without direct contact with the Vistula River could be shaped indirectly by underground water movements when the water level in the river changed.

### 3.2. Physical and Chemical Parameters

The non-metric multidimensional scaling (nMDS) revealed remarkable differences in environmental conditions (SD, $T_W$, DO, EC and pH) of the studied floodplain lakes considering the degree of their connectivity with the river. It allowed us to determine the index of multivariate dispersion of data from individual lakes. Based on environmental conditions nMDS analysis divided the collected samples between the types of floodplain lakes: isolated, transitional and connected (Figure 3).

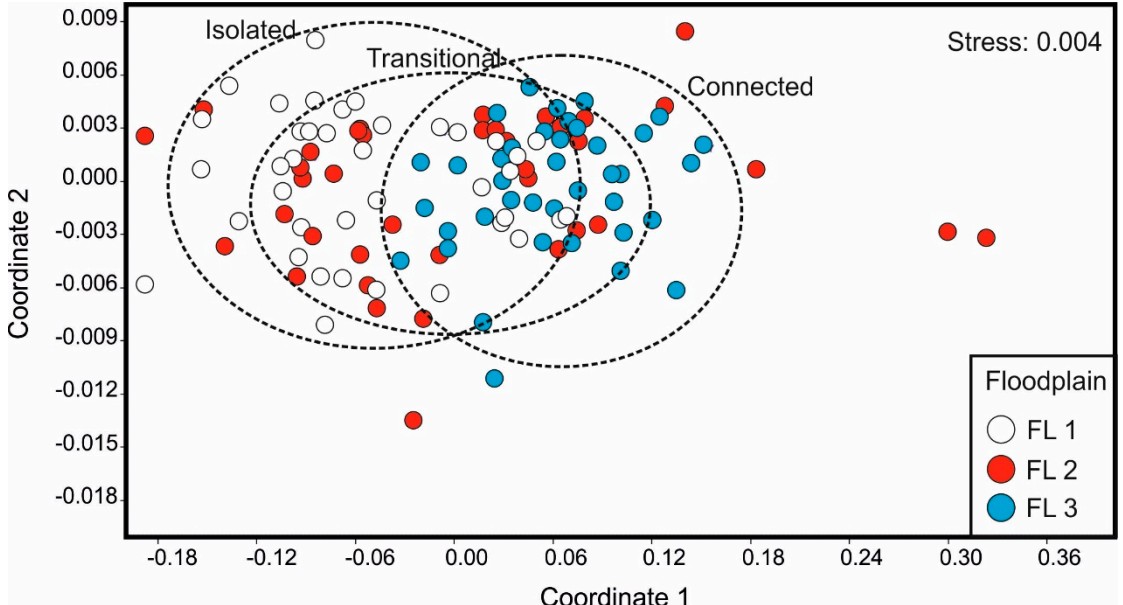

**Figure 3.** Non-metric multidimensional scaling analysis (nMDS) of the environmental conditions (Secchi disk visibility (SD), temperature ($T_W$), dissolved oxygen (DO), conductivity (EC) and pH) (Stress: 0.004) in three types of floodplain lakes (FL1—isolated; FL2—transitional; FL3—connected).

The highest average water transparency (1.45 m) was noted in the isolated lake, and the lowest (0.63 m), in the connected lake ($K$–$W$ = 33.73, $p \leq 0.001$). In contrast, the differences in average temperatures ($T_w$) and conductivity (EC) among the studied floodplain lakes were not statistically significant. The highest average temperature was recorded in the connected lake (20.0 °C), and the lowest, in the transitional lake (19.1 °C). The highest average value of EC was recorded in the connected lake (625), and the lowest, in the isolated lake (576) (Table 2). The highest average concentration of dissolved oxygen in water (DO) was recorded in the connected lake (8.96 mg L$^{-1}$), and the lowest, in the isolated lake (7.87 mg L$^{-1}$) ($K$–$W$ = 10.72, $p$ = 0.01). The highest pH was recorded in the transitional lake (8.6), and the lowest, in the isolated lake (8.3), ($K$–$W$ = 10.97, $p$ = 0.01). Statistically significant differences among sites in visibility (SD), dissolved oxygen (DO) and pH value were observed.

**Table 2.** Water parameters of the investigated floodplain lakes and the Vistula River. Data present the mean (s.d.) for every month samples in 2006–2013. Bold shows significant differences (nonparametric Kruskal–Wallis test, * $p \leq 0.05$; *** $p \leq 0.001$) among water bodies.

| | FL1 (*n* = 34) | FL2 (*n* = 38) | FL3 (*n* = 39) | RV (*n* = 24) |
|---|---|---|---|---|
| Water Temperature (°C) | 19.2 (4.2) | 19.1 (4.1) | 20.0 (3.7) | 19.3 (3.7) |
| **Visibility (cm) *** | 1.45 (0.44) | 0.75 (0.21) | 0.63 (0.23) | 0.54 (0.12) |
| **pH *** | 8.32 (0.36) | 8.60 (0.63) | 8.51 (0.32) | 8.26 (0.22) |
| **Dissolved Oxygen (mg L$^{-1}$) *** | 7.87 (2.72) | 8.29 (2.59) | 8.96 (2.20) | 7.14 (1.66) |
| EC (µS cm$^{-1}$) | 576 (95) | 615 (114) | 625 (84) | 596 (80) |

### 3.3. Taxonomic Richness and Abundances

Samples analysis revealed the presence of 97 zooplankton species in the investigated lakes, including 75 rotifer species (i.e., 77% of all species), 22 crustacean species (i.e., 33% of all species) and nauplii and copepodites, larval forms of Copepoda. The highest number of species (73) was recorded in the transitional lake. The lowest number of species (52) was recorded in the isolated lake (Table 3). There were fewer species (47) in the main channel of the Vistula River (Table 3). The results also indicated that the highest number of species (both rotifers and crustaceans) was recorded in the transitional lake (54 and 19, respectively). The lowest number of species (both rotifers and crustaceans) was recorded in the isolated lake (44 and 8, respectively). The statistically significant differences among the sites were observed in the number of crustacean species ($K–W = 20.30$, $p \leq 0.001$) (Table 3). The list of zooplankton taxa is presented in Table S1 (Supplementary Materials).

**Table 3.** Zooplankton structure of the investigated floodplain lakes and the Vistula River. Data present the mean (s.d.) for every samples between 2006 and 2013, including only dominant taxa (D > 10%). H'—α-diversity, J'—evenness. Significant differences (nonparametric Kruskal–Wallis test; ** $p \leq 0.01$; *** $p \leq 0.001$) among the investigated water bodies are shown in bold.

| | FL1 (*n* = 34) | FL2 (*n* = 38) | FL3 (*n* = 39) | RV (*n* = 24) |
|---|---|---|---|---|
| Richness | 52 | 73 | 63 | 47 |
| **No. of Crustacea species *** | 8 | 19 | 14 | 6 |
| No. of Rotifera species | 44 | 54 | 49 | 41 |
| Total density (ind L$^{-1}$) | 1111 (1255) | 1995 (3406) | 1847 (2587) | 344 (416) |
| H' index | 1.58 (0.47) | 1.80 (0.37) | 1.70 (0.50) | 1.97 (0.27) |
| J' index | 0.581 (0.151) | 0.656 (0.147) | 0.633 (0.153) | 0.745 (0.108) |
| **Crustacea (ind L$^{-1}$) *** | 300 (298) | 290 (329) | 451 (697) | 21 (22) |
| *Bosmina longirostris* | 49 | 67 | 145 | 3 |
| **nauplii *** | 181 | 145 | 189 | 15 |
| **Rotifera (ind L$^{-1}$) ** | 811 (1001) | 1705 (3285) | 1396 (2359) | 323 (407) |
| *Keratella tecta* ** | 21 (33) | 517 (1859) | 295 (632) | 91 (146) |
| *Keratella cochlearis* | 286 (577) | 290 (775) | 114 (190) | 57 (71) |
| *Keratella quadrata* ** | 29 (34) | 74 (173) | 26 (38) | 5 (9) |
| *Polyarthra longiremis* | 308 (509) | 207 (517) | 359 (1584) | 44 (81) |
| *Brachionus angularis* | 5 (8) | 178 (478) | 43 (103) | 25 (49) |

The average zooplankton density in the studied floodplain lakes was 1651 ind L$^{-1}$. The highest average zooplankton density was recorded in the transitional lake (1995 ind L$^{-1}$), and the lowest, in the isolated lake (1111 ind L$^{-1}$) (Figure 4A). The average rotifer density was more than twice as high in the transitional lake as in the isolated one (Table 3, Figure 4A). The difference in the rotifer density among the studied sites was statistically significant ($K–W = 12.27$, $p = 0.007$). The average crustacean density was the highest in the connected lake (451 ind L$^{-1}$) and the lowest in the transitional lake (290 ind L$^{-1}$) (Table 3, Figure 4A). The difference in the crustacean density between the sites was also statistically significant ($K-W = 46.56$, $p \leq 0.0001$). The density of dominant species was the highest in the transitional lake, e.g., *Keratella tecta*, 517 ind L$^{-1}$, *Keratella quadrata*, 74 ind L$^{-1}$ (Table 3). The lowest

*K. tecta* density was noted in the isolated lake, and the lowest *K. quadrata* density, in the connected lake (Table 3, Figure 4B). The differences in the density of *K. tecta* and *K. quadrata* between the studied sites were statistically significant (*K–W* = 12.78 and 15.80, *p* ≤ 0.001, respectively).

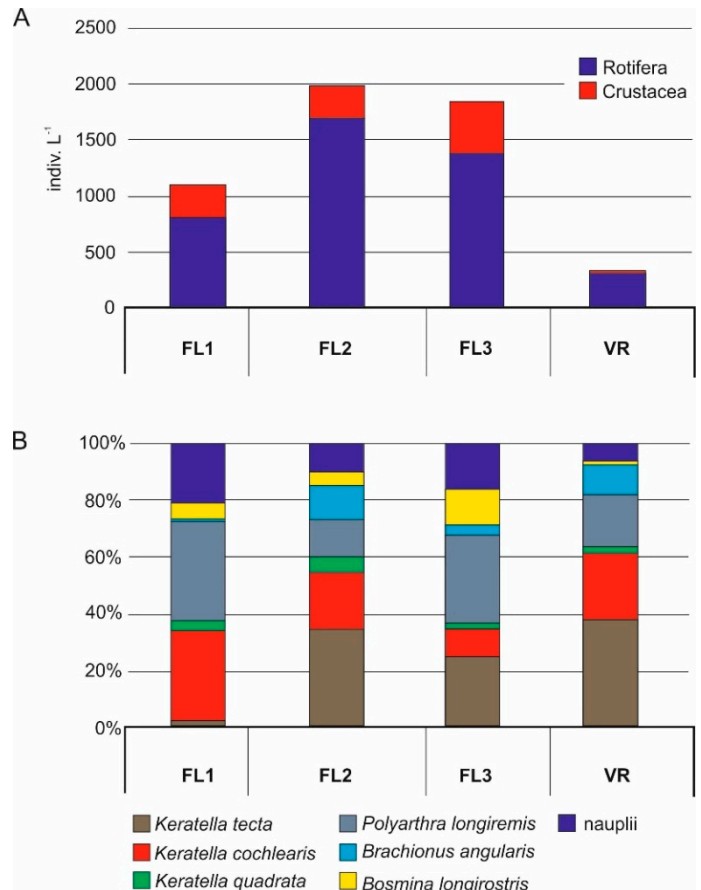

**Figure 4.** Zooplankton density (ind L$^{-1}$) in the investigated floodplain lakes (FL1—isolated, FL2—transitional, FL3—connected) and in the Vistula River (VR) (**A**); Percentage share of dominant taxa in zooplankton density in the studied lakes and in the Vistula River (**B**).

In addition, the average density of *Keratella cochlearis* and *Brachionus angularis* were also the highest in the transitional lake (Table 3). However, for these two species, the differences in the average density among the studied sites were not statistically significant.

The highest density of dominant crustacean forms (e.g., copepod larval forms nauplii) and species was recorded in the connected lake. The lowest density of nauplii was recorded in the transitional lake (*K–W* = 34.47, *p* ≤ 0.0001). *Bosmina longirostris* (Cladocera) was almost three times more abundant in the connected lake than in the isolated one (Table 3). *K. tecta* had the largest share in density among species in the river (VR) and in the transitional lake (Figure 4). *Polyarthra longiremis* and *K. cochlearis* had the biggest share among the dominant species in the isolated lake, while *P. longiremis* and *K. tecta*, in the connected lake. Based on the results it can be concluded that the Vistula River had the highest impact on the transitional and connected floodplain lakes (e.g., *K. tecta*—Dunn's test, *p* ≤ 0.01) (Table 3, Figure 4B). The highest value of α-diversity (H' = 1.80 ± 0.37) and evenness index (J' = 0.656 ± 0.147) were noted in the transitional lake (Table 3) while the lowest, in the isolated lake (α-diversity H' = 1.58 ± 0.47), evenness index (J' = 0.581 ± 0.151).

### 3.4. Influence of Environmental Factors on Zooplankton Communities

The RDA revealed a relationship between zooplankton species composition and environmental variables (Figure 5A). The results of the ordination showed that the eigenvalues of the first ($\lambda_{RDA1}$ = 0.407) and second ($\lambda_{RDA2}$ = 0.169) RDA axes accounted for 57.6% of the variation in the environmental data. All canonical axes were significant (Monte Carlo test, $p$ = 0.002).

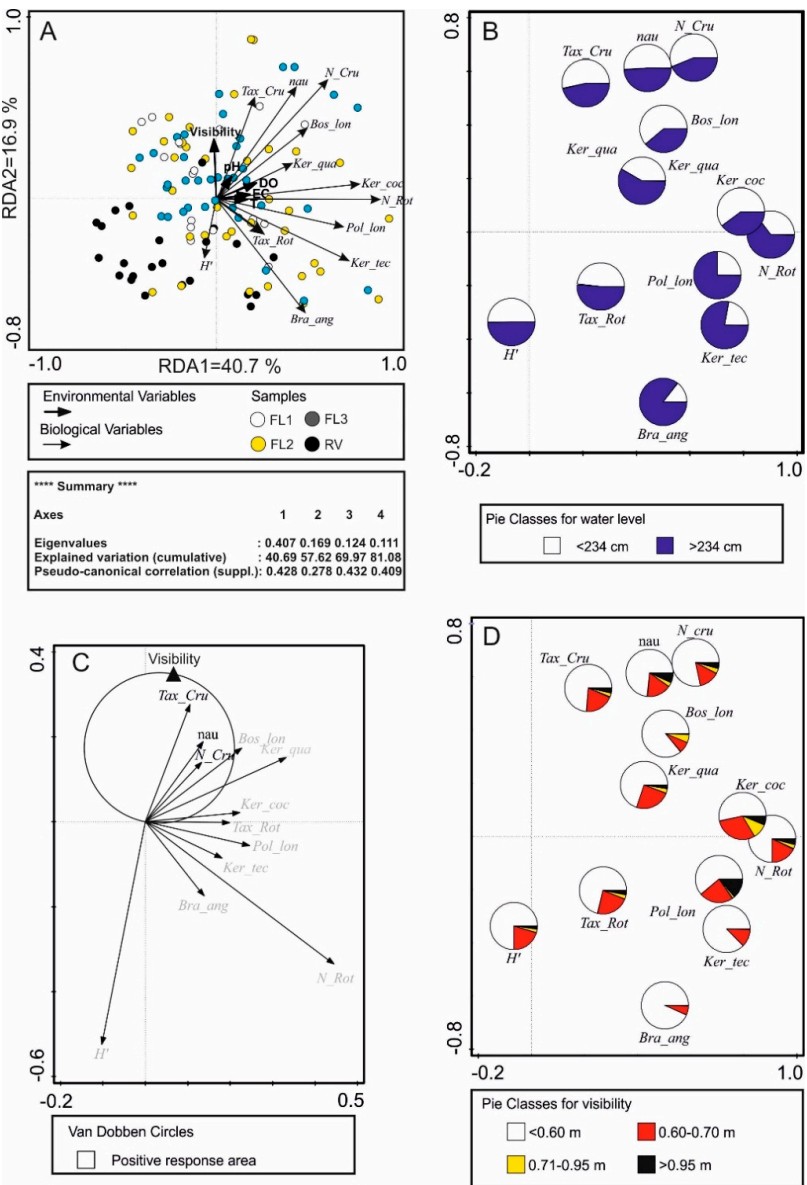

**Figure 5.** Results of redundancy analysis (RDA) performed on zooplankton and environmental data for the three types of floodplain lakes using forward selection of variables ($p$ < 0.05). (**A**) Triplot of significant environmental variabilities, zooplankton and samples; (**B**) relative values of zooplankton communities (pies charts) in relation to the water level in river channel and floodplain lakes (<234 cm—limnophase, >234 cm—potamophase); (**C**) Van Dobben circles analysis (visibility vs. zooplankton structure). A circle indicates positive responses; (**D**) relative values of zooplankton communities in pie charts in relation to visibility. Codes: H'—α-diversity index, Tax_Rot—number of rotifer species, Tax_Crus—number of crustacean species; N_Rot—density rotifers, N_Cru—crustaceans, nau—nauplii, Bos_lon—*Bosmina longirostris*, Ker_qua—*Keratella quadrata*, Ker_coc—*Keratella cochlearis*, Pol_lon—*Polyarthra longiremis*, Ker_tec—*Keratella tecta*, Bra_ang—*Brachionus angularis*; Tw—Temperature, EC—Electrolytic Conductivity, DO—Dissolved Oxygen.

The longest vector describing the visibility is closely correlated with the second axis of ordination. In contrast, the temperature, conductivity and oxygen concentration vectors were well correlated with the first axis of ordination, but were short, which means that these parameters are of less importance. Similarly, the pH vector, which was more closely correlated with the second axis of ordination, was short so that pH was also less important in this analysis.

The RDA indicated that zooplankton did not show a preference for river habitat. Crustacean zooplankton preferred the lake connected with the river, while rotifers the transitional lake (Figure 5A). According to pie charts analysis for water level (Figure 5B) we noted the following relationships: during higher river level (>234 cm) rotifers developed more efficiently, as manifested in higher average density of rotifers, higher average density of *K. tecta*, *B. angularis* and *P. longiremis;* during lower river level (<234 cm) crustaceans developed more efficiently, as manifested in the higher average density of Crustacea as well as in the higher average density of *B. longirostris*. In addition, rapid development of *K. cochlearis* (rotifers) was also observed.

Based on the analysis of RDA and Van Dobben circles (Figure 5C) the visibility was positively correlated with the average number of crustacean species, average density of crustaceans and average density of nauplii, copepod larval forms. Low water transparency was preferred by *K. tecta* and *B. angularis* (indicator species of high trophy) (Figure 5D).

## 4. Discussion

We investigated zooplankton in three floodplain lakes, created as a result of the regulation of the lower Vistula in the mid-19th century. The lakes differed in terms of hydrology, among others in the degree of their connection with the river (isolated, transitional and connected lake). Lateral connectivity of the lakes was responsible for nutrient cycling and biodiversity in the river–floodplain system [1,2,9,15]. The type of connection between the floodplain lake and the river is also important. According to Paira and Drago [5] direct connection occurs when the channel between a floodplain lake and a river is shorter than 1 km. Periodic floods in the Vistula River cause the inclusion of the floodplain lakes into the river system (Figure 2). Massive floods destabilize environmental conditions in floodplain lakes by reducing water transparency, lowering water temperature and inhibiting macrophyte development [39–42]. All these changes have a negative impact on the zooplanktonic population. This observation is true only for intensive flooding. However, on the majority of dates (Figure 2) we recorded medium or small flooding, which did not significantly affect environmental conditions in the studied lakes [20].

The diversity of habitat conditions in floodplain lakes results from several factors including the following: lake–river distance, permanent versus temporary connection between the lake and the river, the size and shape of the lake [2]. The results of our studies, based on environmental conditions, divided floodplain lakes into three types: isolated (FL1), transitional (FL2) and connected (FL3) (Figure 3). The abundance and diversity of invertebrates is generally higher in floodplain lakes compared to the main channel of the river while the taxonomic structure of individual clusters is usually similar [11,43,44]. Zooplankton in the Vistula River was less diverse and less abundant than zooplankton in the studied floodplain lakes because of specific environmental conditions in the river e.g., turbulent water flowing (Figure 4). The taxonomic composition of zooplankton in the Vistula river has impact on the structure of zooplankton communities in the studied floodplain lakes (Figure 4B). A similar relationship has also been observed by other authors [45].

Rotifers predominated in zooplankton diversity and density in the river and floodplain lakes but their predominance varied depending on hydrological conditions. They constituted approximately 90% of zooplankton density in the river, and 60%–80% in the floodplain lakes (depending on the connectivity) (Table 3, Figure 4A). In the lakes, rotifers were (partially) replaced by crustaceans (Figure 4A). According to many authors [18,46,47], Rotifera predominate in both standing and flowing waters owing to their tolerance to changing environmental conditions. Their dominance is believed to be connected with their small size and relatively short development time compared to crustaceans [48–50]. Rotifera

display life history r-strategies and are more adaptable to environmental disturbances of lotic and semi-lotic habitats [18,50].

The greatest mean density of zooplankton and rotifers was recorded in the transitional floodplain lake (Table 3). In this water body rotifers also had the largest percentage share among the dominant species (over 80%) (Figure 4B).

The transitional floodplain lake was temporarily washed out with water from the Vistula: on 19 out of 38 sampling dates the lake was connected to the river. Presumably because of temporary inundation it had the highest number of zooplankton species (both rotifers and crustaceans) (Table 3). The highest mean density of the most dominant species among rotifers and the highest values of H′ and J′ indices were also observed here. The results indicated that it had more diverse zooplankton compared to the isolated and connected lakes.

The gentle washing of the lake at medium water level in the Vistula River might have stimulated the development of zooplankton in accordance with the Intermediate Disturbance Hypothesis [19]: the density and diversity of zooplankton was greater in the lakes connected even temporarily with the main river channel compared to the isolated one [23,24]. At intermediate levels of disturbance (medium flooding during potamophase), species diversity should be greatest because many taxa tolerate existing environmental conditions but none can dominate in the population [51]. Our observations confirmed that the best conditions for zooplankton development were found in the lake periodically connected with the river. According to Mitrovic et al. [52] organic matter supply from inundation could stimulate heterotrophic bacterioplankton and affect zooplankton density and structure. High food availability is related to greater density of zooplankton, which is dominated by small-sized rotifers [53], e.g., *K. tecta* (Table 3, Figure 4B). Rotifers preferred the transitional floodplain lake (Figure 5A).

Based on the percentage share of dominant species (e.g., *K. tecta)* it could be concluded that the river had the greatest impact on zooplankton density in the transitional and connected floodplain lakes (Figure 4B). The differences between the sites were statistically significant. A river can shape the zooplankton community structure in floodplain lakes by periodical flooding [54,55].

The greatest mean density of crustacean zooplankton was observed in the connected floodplain lake, where it constituted approximately 30% of the predominant species density (Figure 4B). The isolated floodplain lake (FL1) did not have the best conditions for the development of crustacean zooplankton: neither its density nor the total number of crustacean species was the highest at this site. On the contrary, the number of crustacean species was the lowest in this lake. A small representation of crustaceans in the isolated floodplain lakes may result from fish pressure [45,46].

On the other hand, it was surprising that the highest density of crustacean zooplankton was recorded in the floodplain lake permanently connected to the river. Presumably, the development of crustaceans was determined by the way in which the lake was connected with the river, i.e., through a narrow channel with the length of 1.2 km and the width of 50 m. With this length the channel could limit or prevent the connection. Moreover, the large size of the lake could stabilize the environmental conditions, which promoted the development of macrophytes (e.g., *Elodeanuttalli*). Owing to that, living conditions for large-sized zooplankton improved significantly. Our study demonstrated that crustacean zooplankton preferred the lake connected with the river (the highest density of crustacean) (Figure 5A). Similar results are obtained by Vadadi-Fülöp et al. [56] for lakes in the Danube valley. A lower degree of connectivity with the river ensures better (more stable) conditions for zooplankton, especially Crustacea. The isolated lake has a well-developed macrophyte community, which serves as a hiding place for zooplankton [57,58]. Cladocera groups, very sensitive to periodic flooding [56], develop faster in isolated or semi-isolated floodplains. Several authors observe that the highest density of zooplankton is recorded in lakes permanently connected to the river by a long channel [17,39,59].

Conditions in floodplain lakes permanently (FL3) or temporarily (FL2) connected to the river depend on flood pulse dynamics (intensity) [47,60–62]. During higher river levels (>234 cm, potamphase), rotifers developed better in the lakes, which was manifested in the higher average density of rotifers and higher density of dominant species, such as *K. tecta*, *B. angularis* and *P. longiremis*

(Figure 5A,B). Higher water level in the Vistula caused an inflow of river water to the lakes (transitional and connected) and facilitated the higher entry of species such as *K. tecta* or *B. angularis* (Figure 4B). The inflow of river water enriched the lakes with river species (mainly rotifers) and this resulted in an increase of the α-diversity index (H′). Such a regularity is also observed by Simões et al. [63]. The river water carried mineral and organic suspension into the connected lakes, thus providing food for detritivorous rotifera, such as *K. tecta* or *B. angularis*. Similar phenomena are observed by Bomfim et al. [53].

Flood pulses also cause nutrient inputs [64], which stimulate phytoplankton production [65,66]. Rapid growth of small edible phytoplankton in floodplain lakes provides good quality food for zooplankton [67–70]. During lower river levels (<234 cm; limnophase), crustacean zooplankton developed better, as noted in the crustacean density and *B. longirostris* density. Lower river levels stabilized environmental conditions not only in the transitional lake (limnophase), but also in the connected lake due to the specific character of the connection (Figure 5B). Periodic flooding causes the succession of two phases in the floodplain lake life cycle: limnophase (isolation) and potamophase (connection) [71]. These two phases differ in water transparency. Water transparency expressed as visibility (SD) was correlated with the average number of crustacean species, average density of crustaceans and average density of nauplii (Figure 5C). Lake isolation promotes the development of Crustacea. Burdis and Hoxmeier [72] note that slower current velocity, lower turbulence and longer residence time are important mechanisms favouring crustacean development in floodplain habitats. Copepods predominated in the crustacean zooplankton of the studied floodplain lakes of the lower Vistula. Their most common forms included larval nauplii (isolated and transitional lake during limnophase), similarly to what is observed in other investigated floodplain lakes [18,65]. During periods with low water transparency rotifer species (*K. tecta* and *B. angularis*) which are indicators of high trophy developed faster [73] (Figure 5D).

It is not easy to answer how, directly, connectivity (dispersal) matters for the persistence and performance of metacommunities [10]. Unfortunately, it is difficult to distinguish between organisms belonging to the adapted vs. dispersed group in zooplankton of floodplain lakes. All pelagic organisms (both alive forms and resting eggs) could be dispersed from river to local communities so it is a very important process. However, local factors such as habitat heterogeneity, water quality, and community interactions can affect the survival and reproduction of individuals [4,13,21,58]. This issue can be explored by studying resting eggs in bottom sediments and interstitial waters. This kind of investigation would help answer the question about the origin of zooplankton in floodplain lakes and dispersion possibilities. We intend to conduct this type of research in future.

Also, it is difficult to compare dispersal probability of zooplankton and settled macroinvertebrates [11] in floodplain-river systems. Based on literature and on our studies Rotifera of the Brachionidae family are best adapted to unstable conditions in floodplain lakes and could be easily dispersed in different water bodies [17,18,39].

## 5. Conclusions

The degree of connection between floodplain lakes and the river affected the zooplankton structure. However, contrary to the initial assumption, the diversity and density of zooplankton were higher in the lake connected with the river (FL3) than in the isolated one (FL1). Zooplankton was the most diverse and abundant in the transitional lake (FL2). The number of crustacean species was also the highest at this site. Regardless of the type of connection, the zooplankton community was less diverse and less abundant in the river than in its floodplain lakes.

The gentle washing of the lake might have stimulated the development of zooplankton in accordance with the IDH theory. The intensity of flood-pulse (inundation) determined a degree of connection between the floodplain lakes and the river and shaped the zooplankton structure (FL2). Higher river level (potamophase) increased zooplankton diversity (higher α-diversity and rotifer density). Lower river level (limnophase) increased crustacean density.

**Supplementary Materials:** The following are available online at http://www.mdpi.com/2073-4441/11/9/1924/s1, Table S1: List of zooplankton taxa at studied floodplain lakes and at the Vistula River.

**Author Contributions:** P.N. conceived and designed the study; P.N., M.B., N.M., M.S. and K.O. performed the field sampling, measurements and data analysis; P.N., K.O. and N.K. wrote and revised the manuscript; P.N. and K.O. provided conceptual overview of the manuscript preparation, writing

**Funding:** This research was funded by the Polish Minister of Science and Higher Education grant number 008/RID/2018/19 by, under the program "Regional Initiative of Excellence" in 2019–2022.

**Conflicts of Interest:** The authors have declared no conflict of interest.

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
