# Peer review of "The Effect of Hydrological Connectivity on the Zooplankton Structure in Floodplain Lakes of a Regulated Large River (the Lower Vistula, Poland)"

_water, doi:10.3390/w11091924_

Round 1

Reviewer 1 Report

In general, the article is well-written, and present a case monitoring study for 3 lakes connected to the Vistula River in different ways.

Since the number of lakes studied cannot be increased easily, I would suggest a little bit more discussion on differences in the potential role of habitat complexity (the three lakes also differ in morphology, size, littoral to pelagic ratio, and so on) between the three lakes. Explaining everything with the IDH hypothesis, therefore, may require caution. I would expect that not only the way these lakes are connected to the river matters, but also the environmental complexity these lakes provide for the (co)occurrence of different zooplankton taxa. Could we understand species richness/composition better as a function of 1) connectivity (dispersal) between river and lake habitats, and habitat complexity (lake morpological data)(e.g.partition the variance explained in community composition)?

I miss some discussions related to general ecological theories, i.e. how connectivity (dispersal) matters for the persistence and performance of (meta) communities. Could zooplankton taxa be group based on functional characteristics, or life forms to better identify their origins (adapted versus dispersed elements) and explain their (co) occurrence in a more mechanistic way?

Author Response

Response to reviewers’ comments
Reviewer 1#

We would like to thank all reviewers for the prompt and thorough revision of our manuscript. All the comments were very constructive; we agreed with almost all suggestions. The text has been carefully reviewed and corrected by a freelance translator who specializes in biological translations.

Since the number of lakes studied cannot be increased easily, I would suggest a little bit more discussion on differences in the potential role of habitat complexity (the three lakes also differ in morphology, size, littoral to pelagic ratio, and so on) between the three lakes. Explaining everything with the IDH hypothesis, therefore, may require caution. I would expect that not only the way these lakes are connected to the river matters, but also the environmental complexity these lakes provide for the (co)occurrence of different zooplankton taxa. Could we understand species richness/composition better as a function of 1) connectivity (dispersal) between river and lake habitats, and habitat complexity (lake morpological data)(e.g.partition the variance explained in community composition)?

Ad. We agree with the Reviewer that discussion on a potential role of habitat complexity in zooplankton development could be interesting. However, although the investigated floodplain lakes differed in size and morphology, the habitat complexity during the vegetation season seemed to be low. It is difficult to discuss environmental complexity when floodplain lakes are shallow or very shallow (mean depth ranging from 0.5 m to 1.5 m) and their water transparency range from 0.6 to 1.5 m. The studied lakes tend to be overgrown with macrophytes and are therefore similar to astatic water bodies. Generally, we observed changes, such as decreased transparency, death of submerged plants and algae and development of floating plants, shortly after a water level rise.

Therefore, according to Amoros and Bornette (2002), hydrological conditions seem to be the most important factor shaping the environment of our floodplains and their zooplankton communities. Hydrological conditions can also affect the import of zooplankton from the river.

We have added the sentence in the chapter Materials and methods, lines 113-114:  “Although floodplain lakes differed in size and morphology, the habitat complexity during the vegetation season seemed to be low.”

I miss some discussions related to general ecological theories, i.e. how connectivity (dispersal) matters for the persistence and performance of (meta) communities. Could zooplankton taxa be group based on functional characteristics, or life forms to better identify their origins (adapted versus dispersed elements) and explain their (co) occurrence in a more mechanistic way?

Ad. Unfortunately, it is difficult to distinguish between organisms belonging to the adapted vs. dispersed group in zooplankton of floodplain lakes. All pelagic organisms (both alive forms and resting eggs) could be dispersed from river to local communities so it is a very important process. However, local factors such as habitat heterogeneity, water quality, and community interactions can affect the survival and reproduction of individuals. (Thomaz et al. 2007, Kobayashi et al. 2015, Karpowicz 2014). This issue can be explored by studying resting eggs in bottom sediments and interstitial waters. This kind of investigation would help answer the question about the origin of zooplankton in floodplain lakes and dispersion possibilities. We intend to conduct this type of research in future.

As Reviewer pointed out “the number of studied lakes cannot be increased easily” so discussion related to general ecological theories may not be sufficiently supported.

Also it is difficult to compare dispersal probability of  zooplankton and settled macroinvertebrates in floodplain - river system. 

Based on literature and on our studies Rotifera of the Brachionidae family are best adapted to unstable conditions in floodplain lakes and could be easily dispersed in different water bodies.

Reviewer 2 Report

Manuscript #: Water-572452

The effect of hydrological connectivity on the zooplankton structure in floodplain lakes of a regulated large river (the lower Vistula, Poland), by Napiorkowski et al.

This study deals with the zooplankton community structure in floodplain lakes in relation to connectivity with the adjacent river. The analyses and results are straightforward and I believe that the findings can contribute to the zooplankton ecology in the light of shaping of zooplankton community structure by hydrological events (e.g., flood pulse) in floodplain lakes.

However, some parts of the manuscript and context of arguments are not clear and awkward, and they need to be clearly revised before publication. Also arguments in many parts are too strong even without direct evidences. Authors need to be more cautious in the expressing the context with to use only “verb” without direct evidence (e.g., The gentle washing “stimulated” the development of zooplankton. --> “might stimulate”)

There are many parts in which only one sentence is composed of one paragraph, which interferes the logical flow of context. The related parts should be gathered in the same paragraph. For example, lines 54-73 had better be organized in one paragraph (there are also such parts in the discussion).

Followings are some concerns and comments which should be considered in the revision.

Line 16 (also see line 344). Clarify “gentle washing” specifically. Does this expression is relevant in the study lakes? The lakes have outlet for the water flow out? Lines 64-72. The authors simply put forward several hypotheses in the last part of Introduction. Have you tested (or speculated) all those hypotheses? Or only showing plausible hypotheses? What is the objective of the paper? Please clarify the goal of the study more clearly in this part. Lines 87-89. What does “n” means? Was that lake number or sample number? Please define it. Line 154 (RDA) and Figure 5. There are no information on the analysis of Fig. 5 (especially 5-B, C, D). What does each of the 5-B (Van Dobben Circles), C, and D (Pie classes) mean? The methods and meaning of those should be included in the corresponding method section or in the legend of the figure. Lines 190-191. Clearly specify “environmental data”. What does “not normal” mean? Please clarify it. Line 196 (Figure 3): Please specify “environmental condition” clearly. Line 223. Does “in the connected lake” correct? Doesn’t it “isolated lake”? Please check it out. Lines 312-313. ”which did not … in the studied lakes.” Where is the evidence on this argument? Please include. Lines 320-321. Please add reference for the part of “while the taxonomy …… is usually similar.” Line 322: …”because of specific environmental condition in the river (Figure 4). Referring Fig. 4 here is relevant? What is “specific” environmental condition? Need to specify. Line 323-324. This part is not clear and awkward. Please rewrite this part. Lines 351-355, 365-366. I advise that the author put some possibilities (food source such as organic matter, bacteria, and phytoplankton; and predator such as fish) on affecting zooplankton community structure other than hydrology-driven factors in a separated paragraph. Line 374. Do you really think that zooplankton “preferred” the connected lake? Don’t you think that they “were preferred” by the connected lake environment? It seems to be chicken-and-egg type reasoning, but this is worthy to speculate. Lines 375-381. This part is contrasting with the results (Table 3, Fig. 4 and lines 367-368). What is the main point of that argument here? Is that relevant to put here? Also there are some confusing arguments (lines 396-399, lines 404-406, line 424) particularly with this result. Although crustaceans were more abundant in the connected lakes (FL 3), you argue that lake isolation and low river level promote the development of Crustacea. This is only valid for the species number, not density. Line 411. Where is the direct evidence of “high level of trophy” such as nutrients and Chl-a? Lines 415-416. There are not direct evidences of turbulent water flow and lower temperature shown in this paper. Thus this part (because of specific … lower temperature, etc,) is not relevant here. In some parts of the paper, position of reference numbers are not in the order (e.g., line 383). Please check with them in the style.

Author Response

Response to reviewers’ comments
Reviewer 2#

The review was very helpful. All the comments were very constructive and we agree with the majority of the suggestions. Hopefully, we have improved the manuscript.

“Some parts of the manuscript and context of arguments are not clear and awkward, and they need to be clearly revised before publication. Also arguments in many parts are too strong even without direct evidences. Authors need to be more cautious in the expressing the context with to use only “verb” without direct evidence (e.g., The gentle washing “stimulated” the development of zooplankton. --> “might stimulate”)”

AdWe have tried to improve the manuscript by avoiding too strong arguments when they are not supported by direct evidence.

There are many parts in which only one sentence is composed of one paragraph, which interferes the logical flow of context. The related parts should be gathered in the same paragraph. For example, lines 54-73 had better be organized in one paragraph (there are also such parts in the discussion).

AdLine 54-73: According to the suggestion we have linked the paragraphs.

Line 16 (also see line 344). Clarify “gentle washing” specifically. Does this expression is relevant in the study lakes? The lakes have outlet for the water flow out?

Ad“gentle washing” – we have used this phrase repeatedly in our previous articles because we believe that this is the best phrase to describe a rather complex phenomenon during water fluctuations in floodplain lakes.

Lines 64-72. The authors simply put forward several hypotheses in the last part of Introduction. Have you tested (or speculated) all those hypotheses? Or only showing plausible hypotheses? What is the objective of the paper? Please clarify the goal of the study more clearly in this part.

AdWe have clarified the objective of the study and added the following sentence:  “The main objective of our study was to answer the question how different hydrological connectivity between a large regulated river and its floodplain lakes can shape zooplankton communities.”  We have tested all the hypothesis during the study (See: Discussion).

Lines 87-89. What does “n” means? Was that lake number or sample number? Please define it.

Ad. We have added an explanation: n – number of samples

Line 154 (RDA) and Figure 5. There are no information on the analysis of Fig. 5 (especially 5-B, C, D). What does each of the 5-B (Van Dobben Circles), C, and D (Pie classes) mean? The methods and meaning of those should be included in the corresponding method section or in the legend of the figure.

Ad. We have added a description to the subsection Data analysis:

To explore significant positive and negative relationships between zooplankton and specific descriptors of physicochemical properties of lake water, a t-value biplots (with van Dobben circles), which approximate the t-values of the regression coefficients of a weighted multiple regression, were generated. The t-value biplots indicated the zooplankton data that reacted to the tested factor to a large degree. 

We have also added the following caption to Figure 5: “A circle indicates positive responses.” and the following caption to the pie charts: – (B) Relative values of zooplankton communities (pies charts) in relation to the water level in river channel and floodplain lakes.; (D) Relative values of zooplankton communities in pies charts in relation to visibility.

Lines 190-191. Clearly specify “environmental data”. What does “not normal” mean? Please clarify it.

Ad. We have changed the sentence to clearly explain the results of nMDS: The Non-metric multidimensional scaling (nMDS) revealed remarkable differences in environmental conditions (SD, TW, DO, EC, and pH) of the studied floodplain lakes considering the degree of their connectivity with the river.”

Line 196 (Figure 3): Please specify “environmental condition” clearly.

AdLine 196 (Figure 3): environmental condition  - SD, TW, DO, EC, and pH (physico-chemical parameters),  it has been added to the caption for Figure 3

Line 223. Does “in the connected lake” correct? Doesn’t it “isolated lake”? Please check it out.

AdWe are sorry, a typing error occurred. The “connected” has been changed to  “isolated”.

Lines 312-313. ”which did not … in the studied lakes.” Where is the evidence on this argument? Please include.

Ad. Evidence is in literature [20]: the studies by Dembowska et al. [20] provide evidence for this observation. The author studied phytoplankton in the same floodplain lakes.

Lines 320-321. Please add reference for the part of “while the taxonomy …… is usually similar.”

Ad. All cited references apply also to the second part of the sentence. It has been corrected.

Line 322: …”because of specific environmental condition in the river (Figure 4). Referring Fig. 4 here is relevant? What is “specific” environmental condition? Need to specify.

Ad. We have added explanation “e.g. turbulent water flowing”.  The turbulent flow of water in the river can significantly affect the structure of zooplankton (leading to lower zooplankton density compared to stagnant water, and to a higher share of rotifers in zooplankton density), which is presented in Figure 4. 

Line 323-324. This part is not clear and awkward. Please rewrite this part.

Ad. The sentence has been changed: “The taxonomic composition of zooplankton in the Vistula river has impact on the structure of zooplankton communities in the studied floodplain lakes.”

Lines 351-355, 365-366. I advise that the author put some possibilities (food source such as organic matter, bacteria, and phytoplankton; and predator such as fish) on affecting zooplankton community structure other than hydrology-driven factors in a separated paragraph.

AdBoth food sources (organic matter, bacteria, or phytoplankton) and the impact of predation were the result of hydrological factors. According to Amoros and Bornette (2002) (“Connectivity and biocomplexity in water bodies of riverine floodplains”)  hydrological factors should be comprehensively combined with ecological factors, such as trophic relationships.

Line 374. Do you really think that zooplankton “preferred” the connected lake? Don’t you think that they “were preferred” by the connected lake environment? It seems to be chicken-and-egg type reasoning, but this is worthy to speculate.

Ad. We believe that a suitable ecological niche must be created first so that a properly adapted organism can develop in it.

Lines 375-381. This part is contrasting with the results (Table 3, Fig. 4 and lines 367-368). What is the main point of that argument here? Is that relevant to put here?

Ad. We realize that the word order without a description may not be entirely clear. That's why minor corrections have been made. We have changed a part of the sentence:

The isolated floodplain lake FL1 did not have the best conditions for the development of crustacean zooplankton: neither its density nor the total number of crustacean species was the highest at this site.” Indeed, at FL1, both the number of species and abundance were not the highest compared to other sites.

However, only the number of crustacean species was the lowest in this lake (FL1). We have corrected this sentence.

Only the highest density and number of crustacean species might testify to the best conditions for their development at floodplain lake.

Also there are some confusing arguments (lines 396-399, lines 404-406, line 424) particularly with this result. Although crustaceans were more abundant in the connected lakes (FL 3), you argue that lake isolation and low river level promote the development of Crustacea. This is only valid for the species number, not density.

Ad. Corrections have been made in this part of the manuscript. Some sentences have been removed and some have been rebuilt. In line 384 we begin the discussion on flood dynamics. Due to the low and medium water level in the river, the impact of flood dynamics could only be recorded in FL2 and FL3. Fig 4 and Fig. 5 apply to this paragraph. Limnophase and potamophase can only be recorded in water bodies that may have connection with the river (Fl 2 and FL 3).

We have added the following sentence: “Conditions in floodplain lakes permanently (FL3) or temporarily (FL2) connected to the river depend on flood pulse dynamics (intensity).”

Line 411. Where is the direct evidence of “high level of trophy” such as nutrients and Chl-a?

Ad. To avoid starting a new topic, we decided to delete the sentence as its contribution to the article seems unimportant.

Lines 415-416. There are not direct evidences of turbulent water flow and lower temperature shown in this paper. Thus this part (because of specific … lower temperature, etc,) is not relevant here.

Ad. The chapter Conclusions has been rewritten.

In some parts of the paper, position of reference numbers are not in the order (e.g., line 383). Please check with them in the style. 

Ad. We have corrected reference numbers.

Round 2

Reviewer 1 Report

I would kindly ask the authors to incorporate the responses to reviewers into the discussion. Therefore, shortly explain in the core text, their thoughts on habitat complexity, dispersal, and functional composition of zooplankton among habitats. These few sentences and references will increase the quality of Discussion, and open towards general ecological questions over the description of community composition, therefore, the present state. 

Author Response

Response to reviewers’ comments
Reviewer 1#

 We would like to thank for the revision of our manuscript.  The review was very helpful and we have improved the manuscript according the suggestions..

 I would kindly ask the authors to incorporate the responses to reviewers into the discussion. Therefore, shortly explain in the core text, their thoughts on habitat complexity, dispersal, and functional composition of zooplankton among habitats. These few sentences and references will increase the quality of Discussion, and open towards general ecological questions over the description of community composition, therefore, the present state. 

Ad.  We try to incorporate the responses to reviewers into the discussion so we put some sentences at the end of discussion.

“It is not easy to answer how, directly, connectivity (dispersal) matters for the persistence and performance of (meta) communities. Unfortunately, it is difficult to distinguish between organisms belonging to the adapted vs. dispersed group in zooplankton of floodplain lakes. All pelagic organisms (both alive forms and resting eggs) could be dispersed from river to local communities so it is a very important process. However, local factors such as habitat heterogeneity, water quality, and community interactions can affect the survival and reproduction of individuals. (Thomaz et al. 2007, Kobayashi et al. 2015, Karpowicz 2014). This issue can be explored by studying resting eggs in bottom sediments and interstitial waters. This kind of investigation would help answer the question about the origin of zooplankton in floodplain lakes and dispersion possibilities. We intend to conduct this type of research in future.

Also it is difficult to compare dispersal probability of  zooplankton and settled macroinvertebrates in floodplain - river system. Based on literature and on our studies Rotifera of the Brachionidae family are best adapted to unstable conditions in floodplain lakes and could be easily dispersed in different water bodies.”

Reviewer 2 Report

The authors considered the reviewer comments seriously in the revision,and whole manuscript came out to be much improved. All the comments were properly responded. 

However, I think it will be better, in terms of the whole format of the paper, that the section of "3. Data analysis" (line 141-172) belongs to "2. Material and method" section. Modification is not mandatory, but it needs to be considered.

Author Response

Response to reviewers’ comments
Reviewer 2#

We would like to thank all reviewer for the prompt and thorough revision of our manuscript.

The authors considered the reviewer comments seriously in the revision, and whole manuscript came out to be much improved. All the comments were properly responded. 

AD. Thank You

However, I think it will be better, in terms of the whole format of the paper, that the section of "3. Data analysis" (line 141-172) belongs to "2. Material and method" section. Modification is not mandatory, but it needs to be considered.

AD. According to the suggestion we have linked the paragraphs 3 Data analysis with 2 Material and method" section. We also change numbers of chapters
